# Development of 3D Cell-Based Fluorescent Reporter Assay for Screening of Drugs Downregulating Telomerase Reverse Transcriptase

**DOI:** 10.3390/bioengineering12040335

**Published:** 2025-03-23

**Authors:** You Li, Fengli Zhang, Zhen Qin, Shang-Tian Yang

**Affiliations:** William G. Lowrie Department of Chemical and Biomolecular Engineering, The Ohio State University, 151 West Woodruff Avenue, Columbus, OH 43210, USA; li.9334@osu.edu (Y.L.);

**Keywords:** breast cancer, drug discovery, fluorescent reporter assay, high-throughput screening, telomerase reverse transcriptase

## Abstract

A fluorescent cell-based assay was developed for the screening of chemicals repressing the expression of human telomerase reverse transcriptase (hTERT). hTERT is reactivated during carcinogenesis and is overexpressed in more than 90% of cancers but is almost silent in normal tissue cells. Because of its critical role in cancer, hTERT is a target in various therapeutic strategies for cancer treatment. In this study, the hTERT promoter was cloned in MCF7 breast cancer cells and used to control the expression of enhanced green fluorescent protein (EGFP). The fluorescence of EGFP indicated the activity of the hTERT promoter, and, in the presence of an hTERT repressor, the EGFP fluorescence signal was reduced as compared to the EGFP fluorescence controlled by the human cytomegalovirus (CMV) promoter, which was not affected by changes in culture conditions and worked as a control. The EGFP reporter cells were cultivated in three-dimensional (3D) microbioreactors to resemble the in vivo tumor physiology and provide in vivo-like responses. The assay’s predictability was demonstrated with three known hTERT inhibitors, pristimerin, epigallocatechin gallate, and n-butylidenephthalide, and further evaluated with five widely used anticancer compounds, doxorubicin, cisplatin, paclitaxel, blasticidin, and tamoxifen. The results showed overall accuracy of over 83.3%, demonstrating the feasibility of using the hTERT promoter with EGFP as a reporter for the screening of potential cancer drugs targeting hTERT.

## 1. Introduction

Telomerase reverse transcriptase (TERT) is an important subunit of telomerase that synthesizes TTAGGG repeats to telomeres, i.e., the ends of chromosomes [1,2]. Telomeres protect chromosomes against degradation and fusion and preserve genome integrity. Since telomerase activity is negligible after cell differentiation, telomeres in somatic cells continue to shorten with each successive cell division, leading to cell senescence eventually [3,4]. Telomerase activity is restored during carcinogenesis. hTERT is upregulated in approximately 90% of human cancers, which controls telomerase’s lengthening function and contributes to cancer cell immortalization [5]. Many mechanisms for hTERT upregulation have been reported in cancer cells, including transcriptional regulation, mutation, hypermethylation, and alternative splicing [6]. In addition to its canonical function of telomere elongation, hTERT has been reported to affect mitochondrion function and reduce cellular reactive oxygen species (ROS) to prevent cell apoptosis [7].

Among all possible mechanisms, regulation at the transcription level is considered the primary one and has received extensive research interest [8]. Several research groups have deciphered the promoter region of hTERT. Horikawa et al. (1999) discovered that a 59 bp region (−208 to −150 relative to the transcriptional start site (TSS)) containing one E-box site was required for the maximum promoter activity [9]. Wick et al. (1999) found that −43 to −1125 upstream of the start codon allowed the high expression of hTERT [10]. Takakura et al. (1999) found that the activity of hTERT was controlled by a 180 bp region upstream of TSS, in which SP1 and E-box binding sites played important roles [11]. Cong (1999) observed that a region 330 bp upstream of the start codon to the second exon governed the promoter’s activity [12]. Although these studies reveal that the hTERT promoter region has a variety of transcriptional factor binding sites spanning up to 3000 bp, it is concluded that about 330 bp upstream of the start codon could be regarded as the core region of the hTERT promoter associated with hTERT’s activation in cancer cells.

Since TERT is specifically and highly expressed in cancer cells, hTERT has been a therapeutic target for cancer drug discovery and development [13,14]. By targeting the TERT promoter, several reporter gene assays that connect the promoter fragment to a reporter gene to discover hTERT promoter inhibitors have been shown to be effective and extremely useful. Li et al. developed a firefly luciferase reporter controlled by the hTERT promoter to screen 400 chemosynthetic compounds and found that 1-(4-chlorobenzyl)-3-(hydroxyimino)-4-(4-(trifluoromethyl)phenyl)indolin-2-one efficiently suppressed hTERT expression and induced cell apoptosis through a mitochondrial-regulated pathway, demonstrating the antitumor potential of this compound and the effectiveness of using hTERT promoter reporter gene assays for inhibitor screening [15]. Huang et al. constructed another hTERT reporter gene assay where an approximately 3.4k bp hTERT promoter was linked to a reporter gene encoding GFP or secreted alkaline phosphatase (SEAP). They found that SEAP had better performance than GFP, as the fluorescence measurements were affected by the temperature and pH and by background noise from the plates and culture medium [16]. Using the SEAP assay, their group found two tea polyphenols, (K)-epigallocatechin (EGC) and (K)-epigallocatechin-3-gallate (EGCG), as significant hTERT repressors [17]. Luciferase, SEAP, and fluorescent proteins are common reporters used in reporter gene assays. Enzyme-based detection is more sensitive than fluorescent proteins, but it is compromised by disturbing the cell culture system, as external substrate addition is needed pre-assay. In contrast, fluorescent proteins are much more user-friendly because they can be directly visualized or measured by using fluorescent microscopes or plate readers in real time, leaving the culture system undisturbed [18].

The application of 3D culture improves drug toxicity prediction for drug screening assays. In general, 3D cultures can better represent the real tumor microenvironment, including cell–cell and cell–matrix interactions, as well as their effects on cell responses to drugs; they thus give drug toxicity results that are closer to those in vivo [19,20]. In our previous work, we developed a polyethylene terephthalate (PET)-based 3D culture and showed that cells growing on the PET scaffold exhibited an in vivo-like morphology [21]. The scaffold, embedded in a modified 384-well plate, allowed for the easy detection of red and green fluorescent proteins from engineered reporter cells using a plate reader. We also found that this 3D culture greatly improved the fluorescent signal readings due to the higher cell culture density and the 3D scaffold’s fluorescence-focusing effect. This 3D assay has been successfully applied for the screening of survivin gene inhibitors [22,23,24].

In this study, we aimed to develop an hTERT promoter reporter gene assay using EGFP as the reporter protein in cancer cells. Two hTERT promoter fragments (1500 bp and 330 bp) were cloned to drive the expression of EGFP, and the constructs were transfected into multiple cancer cell lines. The assay was performed in a 3D culture to simulate the in vivo tumor microenvironment, which gave more accurate cell responses. The assay was first evaluated with three known hTERT promoter inhibitors, pristimerin, epigallocatechin gallate (EGCG), and n-butylidenephthalide (BP). Then, the assay was further evaluated for its reportability using five anticancer drugs, doxorubicin, cisplatin, paclitaxel, blasticidin, and tamoxifen. The 3D EGFP reporter assay results were then examined by comparing them with the results from mRNA tests. Only tamoxifen showed inconsistent results between the two methods. The accuracy rate of the developed assay was over 80%, demonstrating the potential of this 3D fluorescent reporter assay for early-stage hTERT suppressor identification.

## 2. Materials and Methods

### 2.1. Drug Reagents

Pristimerin, EGCG, BP, doxorubicin, cisplatin, paclitaxel, blasticidin, and tamoxifen were purchased from Sigma-Aldrich (St. Louis, MO, USA). Unless otherwise noted, pristine, epigallocatechin gallate, doxorubicin, cisplatin, and paclitaxel were dissolved in dimethyl sulfoxide; blasticidin was dissolved in the culture medium; and tamoxifen was dissolved in ethanol. Drug stock solutions were stored at −20 °C before being diluted to the various concentrations used in this study.

### 2.2. Plasmid Construction

The plasmid pEGFP-N3 (renamed as pCMV-EGFP) was purchased from Takara Bio (Mountain View, CA, USA). A long hTERT promoter fragment was obtained from lung cancer (A549) cells’ genomic DNA by PCR amplification using the primers 5′-GCAGAGAACCAGTGTAAGC-3′ and 5′-TGTCGCCTGAGGAGTAGA-3′. The obtained promoter fragment was connected to the pCMV-EGFP backbone by Gibson assembly with the replacement of the CMV promoter. The primers used in the Gibson assembly for the construction of plasmids p1500 bp-hTERT-EGFP and p330 bp-hTERT-EGFP are listed below, and the obtained plasmids were verified by the DNA sequencing of the whole plasmids.
**p1500 bp-hTERT-EGFP****p330 bp-hTERT-EGFP**5′-CCCCGCGTCAAGCTTCGAATTCTGC-3′5′-CCCCGCGTCAAGCTTCGAATTCTGC-3′5′-AAGGGGCGGAACTAATGCATGGCGGTAATAC-3′5′-CGGACCCGGGAACTAATGCATGG-3′5′-GCATTAGTTCCGCCCCTTTGCCCTAG-3′5′-AAGCTTGACGCGGGGGTGGC-3′5′-AAGCTTGACGCGGGGGTGGC-3′5′-TGCATTAGTTCCCGGGTCCGCC-3′

### 2.3. Cell Lines and Cell Culture

Human breast cancer MCF7 cells were cultured in T25 or T75 flasks with Dulbecco’s Modified Eagle Medium (DMEM, Sigma) supplemented with 1% (*v*/*v*) non-essential amino acids (Gibco, Invitrogen, Carlsbad, CA, USA), 1% (*v*/*v*) penicillin–streptomycin (Gibco), and 10% (*v*/*v*) fetal bovine serum (FBS, Sigma). Cells cultures were conducted in a 37 °C humidified incubator with 5% CO_2_. Cells were passaged when the cell confluency reached around 70%. Cells were treated with 0.05% trypsin–EDTA (Gibco) for detachment and reseeded for routine cell passage or different experiments.

### 2.4. Generation of Stable EGFP Reporting Cells

Cells were seeded in a 24-well plate for 24 h prior to transfection to achieve cell confluency of 70–80%. The pCMV-EGFP, p1500-bp hTERT-EGFP, and p330-bp hTERT-EGFP plasmids were introduced into the cells using lipofectamine 3000 (Invitrogen), following the manufacturer’s instructions. On the next day, green fluorescence was detected to compare the activity of various promoters. To establish stable EGFP-expressing clones, cells containing the CMV promoter or 330 bp-hTERT promoter reporting cassettes were selected in G418-containing medium for 2 weeks and then expanded in G418-free medium for another 3 to 4 weeks. Following this, single cells with strong green fluorescence were sorted using fluorescence-activated cell sorting (FACS) (BD Biosciences, Woburn, MA, USA) and cultured in 96-well plates with 200 μL medium for single-clone cell expansion. The resulting clones were named MCF7-CMV-EGFP, A549-CMV-EGFP, PANC1-CMV-EGFP, MCF7-hTERT-EGFP, A549-hTERT-EGFP, and PANC1-hTERT-EGFP for MCF7, A549, and PANC1 cancer cells, respectively.

### 2.5. Real-Time PCR for mRNA Quantification

The engineered reporting cells were inoculated in 6-well plates. Drugs (1.5 µM Pristimerin, 200 µM EGCG, or 80 µg/mL BP) were added to the culture at 24 h. After 48 h, cells were collected, and the total RNA was extracted with the RNeasy Mini Kit (Qiagen). The concentration and purity of the obtained RNA were determined by NanoDrop (Thermo Fisher Scientific, Waltham, MA USA). High-Capacity cDNA Reverse Transcription Kits (Applied Biosystems, Waltham, MA USA) were used to reverse-transcribe one microgram of RNA using the primer pairs listed below.
hTERT primers5′-CGGAAGAGTGTCTGGAGCAA-3′ 5′-GGATGAAGCGGAGTCTGGA-3′EGFP primers5′-GCTGACCCTGAAGTTCATCTG-3′ 5′-CACCTTGATGCCGTTCTTCT-3′β-actin primers5′-CAGGTCATCACCATTGGCAATGAGC-3′ 5′-CGGATGTCCACGTCACACTTCATGA-3′

The quantitative real-time PCR (qRT-PCR) reactions were conducted in triplicate using PowerUp SYBR Green Master Mix (Applied Biosystems). The levels of hTERT and EGFP gene expression were normalized to β-actin mRNA and evaluated by 2^−ΔΔCT^.

### 2.6. Fluorescence Imaging

MCF7-CMV-EGFP and MCF7-hTERT-EGFP cells were inoculated into 24-well plates and treated with pristimerin at a predetermined concentration of 0.75 µM for 96 h. Before the fluorescence imaging, 4% paraformaldehyde was applied for 10 min to fix the cells. Subsequently, Hoechst 33342 (Thermo Fisher Scientific) at a concentration of 5 µg/mL was used to stain the cell nuclei for another 10 min. Cell fluorescence from the expressed EGFP (green) and the nuclei stained by Hoechst 33342 (blue) was examined with a Nikon fluorescence microscope under the green and blue fluorescence channels.

### 2.7. Two-Dimensional Cell Culture

MCF7-hTERT-EGFP and MCF7-CMV-EGFP cells were cultured in 6-well plates with an initial density of 150,000 cells/well to study and compare their growth kinetics. The cells in the wells were trypsinized and collected daily, and the cell numbers were determined after trypan blue staining and using a hemocytometer.

### 2.8. Three-Dimensional Cell Culture and Fluorescence Intensity Reading

A non-woven polyethylene terephthalate (PET) scaffold was used to construct the 3D cell culture architecture. The PET was punched into small discs with a 4 mm diameter and 1 mm thickness. To improve cell attachment, the PET scaffolds were pretreated with FBS for over 24 h. Following this, scaffold discs were placed at the center of a 40-microbioreactor (40-MBR) plate, which was redesigned from a black/clear-bottom 384-well plate (BD OptiluxTM, San Jose, CA, USA) as described previously [21,24]. Then, each scaffold was seeded with 20,000 cells (in 15 mL) and placed in a 37 °C incubator for cell attachment onto the scaffold. After static incubation for 4 h, 600 mL of culture medium was added, and the 3D culture was statically incubated for another 24 h before being transferred to a shaker at a rotation speed of 90 rpm for the remaining culturing period. The cell fluorescence was measured daily using a plate reader (GENios-Pro^TM^, Tecan, Männedorf, Switzerland) at 485 nm excitation and 535 nm emission wavelengths for EGFP. The cell fluorescence (F_cell_) was calculated by subtracting the PET scaffold fluorescence (F_scaffold_) determined before cell inoculation and the average medium fluorescence of the neighboring wells (F_medium_) from the total fluorescence of the center well (F_total_).

### 2.9. Three-Dimensional Cytotoxicity Assay

MCF7-CMV-EGFP fluorescent cells were seeded at a density of 20,000 cells for each microbioreactor on the modified multi-well plate. At 48 h post-incubation, culture media were replaced with fresh media containing a drug at a specified concentration (pristimerin: 0, 0.75, 1, 1.5, and 2 µM; doxorubicin: 0, 1, 1.5, and 2 µM; cisplatin: 0, 25, 50, 75, and 100 µM; paclitaxel: 0, 75, 150, 300, and 600 nM; blasticidin: 0, 25, 50, and 100 µg/mL; tamoxifen: 0, 25, 50, 75, and 100 µM). The non-specific cell response or drug cytotoxicity was determined daily for 5 to 6 days by measuring the EGFP green fluorescence intensity. The measured cell fluorescence data were normalized with the initial seeding fluorescence intensity and used in the analysis.

### 2.10. Three-Dimensional Reporter Assay of hTERT Downregulation by Drugs

In parallel to the cytotoxicity assay with MCF7-CMV-EGFP cells, MCF7-hTERT-EGFP reporter cells were also cultured in the 3D microbioreactors to determine the effects of the drugs in downregulating the hTERT promoter. The drugs were added at 48 h, as with the drug cytotoxicity study. EGFP fluorescence intensities were measured daily. Specific hTERT expression was estimated by normalizing the green fluorescence intensity of MCF7-hTERT-EGFP cells to that of MCF7-CMV-EGFP cells. The effects of the drugs on hTERT expression were then determined by the relative hTERT expression or the ratio between the specific hTERT expression with drug treatment to that without drug treatment.

### 2.11. Logistic Regression

Logistic regression was used to develop a predictive model for the classification of the drugs into two categories: hTERT repressors (1) and non-repressors (0). The relative hTERT expression in cells treated with pristimerin, blasticidin, cisplatin, doxorubicin, and paclitaxel in the 3D reporter assay was collected as the training dataset. Multiple features were evaluated in the regression, and the one with 100% prediction accuracy for the training dataset was selected for the prediction model.

### 2.12. Statistical Analysis

Experimental data from mRNA quantification and the 3D assay are expressed as the mean value ± standard error of the mean (*n* ≥ 3). Three triplicates and five triplicates were conducted in mRNA quantification and the 3D assay, respectively. Student’s *t*-test was used, and *p* < 0.05 was deemed statistically significant.

## 3. Results

### 3.1. Characterization of Fluorescent Reporting Cell Lines

MCF7 cells were transfected with fluorescent reporting plasmids CMV-EGFP, 1500 bp-hTERT-EGFP, and 330 bp-hTERT-EGFP, respectively, and the resulting cells were examined for green fluorescence on the following day after transfection. As can be seen in Figure 1A, cells transfected with the CMV-EGFP plasmid emitted strong green fluorescence, whereas only a few cells were green-positive after receiving the 1500 bp-hTERT-EGFP or 330 bp-hTERT-EGFP plasmids. Flowcytometry analysis showed that the EGFP expression was significantly higher under the control of the CMV promoter than the hTERT promoter. FACS was then used to obtain single clones of cells with stronger fluorescence. A strong linear correlation (R^2^ > 0.99) between the EGFP fluorescence intensity (RFU) and the cell number was observed for both MCF7-CMV-EGFP and MCF7-hTERT-EGFP cells (Figure 1B), suggesting that the green fluorescence could be used to monitor changes in cell number in the culture. However, the hTERT promoter was ~20-fold weaker than the CMV promoter, as indicated by the much smaller proportionality coefficient in the correlations.

The growth kinetics of MCF7-CMV-EGFP and MCF7-hTERT-EGFP cells in 6-well plate cultures were monitored by counting the cells daily. Figure 1C shows that the MCF7-CMV-EGFP and MCF7-hTERT-EGFP cells had similar growth kinetics, indicating that EGFP expression did not significantly affect the cell growth behavior.

To demonstrate that the expression level of EGFP could sensitively indicate the downregulation of endogenous hTERT, the mRNA levels of hTERT and EGFP from MCF7-hTERT-EGFP cells were quantified before and after exposure to three known hTERT suppressors, pristimerin, epigallocatechin gallate (EGCG), and n-butylidenephthalide (BP), respectively. As shown in Figure 2, all three drugs downregulated both hTERT and EGFP expression in MCF7-hTERT-EGFP cells, suggesting that the EGFP fluorescence from the MCF7-hTERT-EGFP cells can be used as a reporter to reflect the expression level of endogenous hTERT.

Pristimerin, a methyl ester of celastrol derived from the families of *Celastraceae* and *Hippocrateaceae*, has wide pharmacological application as an anticancer, anti-inflammatory, and antibacterial agent. Its anticancer effect has been observed in a wide range of cancers, including breast cancer [25], lung cancer [26], and pancreatic cancer [27], involving a number of signaling pathways, such as MAPK, PI3K/AKT/mTOR, NF-κB, and Wnt/β-Catenin [28]. It was reported that the downregulation of hTERT was attributed to the suppression of transcriptional factors c-Myc, SP1, STAT3, and protein kinase B/Akt in prostate cancer cells [29] and to both epigenic and genetic regulations in Panc-1 cells [30]. EGCG, a polyphenol compound isolated from green tea, was considered the key chemical component responsible for the anticancer effect of green tea [31]. An EGCG nanoemulsion was found to inhibit lung cancer growth, migration, and invasion via the activation of the AMPK signaling pathway [32]. EGCG was also reported to inhibit Panc-1 cell growth through the PI3K/AKT/mTOR signaling pathways [33]. Regarding the specific effect of EGCG on hTERT, both epigenic modification (promoter methylation and histone acetylation) and genetic modification (repressor E2F-1 binding) were reported in MCF7 cells [34]. BP, a natural compound derived from *Angelica sinensis* (female ginseng), is used clinically to treat gynecological diseases. hTERT expression was found to be attenuated in BP-treated A549 cells, along with the downregulation of hTERT promoter transcription factor AP-2α [35]. BP is hydrophobic and difficult to dissolve in aqueous media. EGCG gave an intensive red color in solution that interfered with the fluorescence reading. Pristimerin was thus chosen to validate the reporting cell pair in the hTERT repressor assay for the screening of the expression level of endogenous hTERT in MCF7 cells.

### 3.2. Validation of Reporting Cell Pair with Pristimerin

To evaluate the drug’s effect on the hTERT promoter, MCF7-hTERT-EGFP cells as the reporters were paired with MCF7-CMV-EGFP cells as the control. These two EGFP cell lines, with different promoters but similar growth kinetics (see Figure 1C), should have similar responses to drug cytotoxicity, as observed in our previous studies [24,36]. MCF7-CMV-EGFP and MCF7-hTERT-EGFP cells were treated with 0.75 μM pristimerin, followed by the analysis of the mRNA levels for hTERT and EGFP in these cells. As expected, the endogenous hTERT expression decreased significantly for both cells, whereas the EGFP expression remained unaffected for MCF7-CMV-EGFP cells but decreased significantly for MCF7-hTERT-EGFP cells (Figure 3A). Fluorescent microscopy images showed that 0.75 μM pristimerin did not affect the EGFP expression in MCF7-CMV-EGFP cells but significantly decreased the EGFP expression in MCF7-hTERT-EGFP cells (Figure 3B). Clearly, 0.75 μM pristimerin was inhibitory to the hTERT promoter but not the CMV promoter, confirming that pristimerin at 0.75 μM downregulated hTERT expression but was not cytotoxic to MCF cells. The EGFP fluorescence of MCF7-hTERT-EGFP cells would be proportional to the hTERT expression level, whereas the EGFP fluorescence of MCF7-CMV-EGFP cells would be proportional to the total cell number or density. Their ratio represents the specific hTERT expression level in cells and thus can be used as indicative of the drug’s effect in downregulating hTERT expression in cells. Therefore, the EGFP reporter cell pair can be used to screen hTERT repressors by comparing changes in the green fluorescence intensities of MCF7-hTERT-EGFP and MCF7-CMV-EGFP cells with and without drug treatment.

### 3.3. Three-Dimensional Reporter Assay for Screening of hTERT Repressors

To establish a 3D reporter assay for the screening of hTERT repressors or inhibitors, pristimerin was used to further evaluate its effects on MCF7 reporter cells cultured in the 3D PET scaffold. The cytotoxicity of pristimerin was first assessed at various concentrations from 0 to 2 μM using the green fluorescence of MCF7-CMV-EGFP cells, which showed dose-dependent responses (Figure 4A). The drug’s cytotoxicity was reflected by reduced cell growth or increased cell death in the presence of the drug, as indicated by the green fluorescence intensities of CMV cells. The drug’s IC_50_ concentration, or the concentration at which the cell number or culture fluorescence was reduced by 50% compared to the control without the drug at the same time point, was approximately 1 µM for pristimerin. We then investigated the downregulation effect of pristimerin at 1 µM on hTERT in the 3D cultures of MCF7-hTERT-EGFP and MCF7-CMV-EGFP cells. As can be seen in Figure 4B, after the drug’s addition at 48 h, cell growth slowed significantly, as reflected by a slower increase in the culture’s fluorescence for both MCF7-CMV-EGFP and MCF7-hTERT-EGFP cells compared to the cultures without the drug. To evaluate the drug’s effect in downregulating the hTERT promoter, the relative hTERT expression was estimated from [NF-hTERT/NF-CMV]_with drug_/[NF-hTERT/NF-CMV]_without drug_, where NF-hTERT is the normalized green fluorescence intensity of MCF7-hTERT-EGFP and NF-CMV is the normalized green fluorescence intensity of MCF7-CMV-EGFP. As shown in Figure 4B, the relative hTERT expression was approximately one in the absence of the drug but decreased significantly to below one after drug addition due to the downregulation of the hTERT promoter by pristimerin, which was confirmed by the mRNA level of hTERT in MCF7 cells with and without drug treatment (Figure 3).

### 3.4. Investigation of Drug Cytotoxicity and hTERT Repressors with 3D Reporter Assay

Next, the 3D assay was used to assess the cytotoxicity and hTERT promoter downregulation of several chemicals with known anticancer and anti-inflammatory activity. Figure 5 shows the cytotoxicity and hTERT downregulation for doxorubicin, cisplatin, and blasticidin. Both doxorubicin and cisplatin are anticancer chemotherapy drugs that act through general DNA damage. Doxorubicin is an antibiotic that harms DNA by intercalating into DNA–DNA pairs and inhibiting topoisomerase II [37]. Cisplatin is a metal coordination compound that acts by crosslinking with the urine bases, preventing DNA repair and thus causing DNA damage in cancer cells [38]. Blasticidin, typically used as a cell selection antibiotic for mammalian and bacterial cells [39], induces cell death through blocking protein synthesis by binding to the ribosomal P-site [40]. As expected, cell growth was inhibited with an increased death rate upon increasing the drug concentration in the range studied (0–2 μM doxorubicin, 0–100 μM cisplatin, 0–100 μg/mL blasticidin) (Figure 5A,C,E). The significant downregulation of hTERT in the MCF7 cells treated with these drugs was also observed, as the relative hTERT expression decreased by ~10% with doxorubicin at 2 μM (Figure 5B), ~18% with cisplatin at 25 μM (Figure 5D), and ~16% with blasticidin at 50 μg/mL (Figure 5F). The downregulation of hTERT in the MCF7 cells was confirmed by the hTERT mRNA levels in the cells, which were reduced to 58% with doxorubicin, 33% with cisplatin, and 25% with blasticidin compared to the control without drug treatment.

We further studied paclitaxel and tamoxifen for their cytotoxicity and potential effects on hTERT expression in MCF7 cells (Figure 6). Paclitaxel is a widely used chemotherapeutic agent because it specifically targets microtubules and arrests cells in the G2/M phase of the cell cycle [41]. However, no drug-dependent responses were observed with paclitaxel at 0–600 nM in the 3D culture (Figure 6A). Paclitaxel at 50 nM was found to be highly toxic and greatly altered the cell morphology in 2D cultures, but these damaged cells remained attached to the bottom surface of the multi-well plate. The damaged cells emitting green fluorescence were likely trapped in the 3D scaffold and thus only a small decrease in the culture fluorescence was observed in the 3D culture treated with 600 nM paclitaxel (Figure 6B). Moreover, paclitaxel at 600 nM did not downregulate hTERT as the relative hTERT expression level remained at ~1.0 and the hTERT mRNA level in cells was not significantly reduced as compared to the control. Earlier studies also reported that paclitaxel had no impact on hTERT [42,43].

Tamoxifen is currently used to treat estrogen receptor-positive breast cancer because it specifically inhibits estrogen’s binding to its receptor [44]. Tamoxifen at 50–100 μM exhibited strong cytotoxicity to MCF7 cells (Figure 6C). With 25 μM tamoxifen, the relative hTERT expression decreased by ~10% in the first 24 h after drug addition but then increased by ~110% on the last day (Figure 6D). The effect of tamoxifen on hTERT expression in MCF7 cells was thus ambiguous and undetermined in the 3D reporter assay.

### 3.5. Validation of 3D Assay with qRT-PCR

To validate the accuracy of the 3D reporter assay, the hTERT mRNAs in the MCF7 cells treated with 1.5 µM pristimerin, 1.5 µM doxorubicin, 50 µM cisplatin, 75 nM paclitaxel, 30 µg/mL blasticidin, and 25 µM tamoxifen were quantified with qRT-PCR. As can be seen in Figure 7, MCF7 cells treated with pristimerin, blasticidin, cisplatin, doxorubicin, and tamoxifen showed significantly decreased hTERT mRNA levels, whereas cells treated with paclitaxel showed no significant differences compared to the control without drug treatment. The decreased hTERT mRNA levels were consistent with the observed hTERT downregulation effects in the 3D reporter assay for pristimerin, blasticidin, paclitaxel, cisplatin, and doxorubicin, but not for tamoxifen. Previous studies have reported that the treatment of MCF7 cells with tamoxifen resulted in hTERT repression and cell inhibition [45,46]. Depending on the cell type, tamoxifen could inhibit or promote cell growth and repress or activate hTERT, possibly resulting from its dual regulation of the two estrogen-responsive elements (ERE) on the hTERT promoter. The EREs are located at around −2700 and −970 upstream of the promoter region [47], and they were not present in the (330 bp)-hTERT-EGFP reporting cassettes used in the 3D reporter assay. This might have contributed to the discrepancy in the results of the 3D reporter assay and mRNA test. Another possible reason could be the differences in the culture environment used in the 3D reporter assay (3D culture) versus the 2D culture to incubate the cells collected for the mRNA assay. As discussed earlier, 3D cultures can better represent the real tumor microenvironment, including cell–cell and cell–matrix interactions, as well as their effects on cells’ responses to drugs; thus, they give drug toxicity results that are closer to those in vivo [19,20]. This remains to be investigated in the future.

### 3.6. Binary Logistic Regression Model

In the 3D reporter assay, the reduction in the relative hTERT expression due to the drug’s inhibitory effect on the hTERT promoter was less than 20%. A binary logistic regression was thus applied to the data from pristimerin, blasticidin, cisplatin, doxorubicin, and paclitaxel to establish a model for the classification of the drugs in terms of their effects on hTERT into two categories—hTERT repressors and non-repressors. Table 1 shows the relative hTERT expression in MCF7 cells treated with various drugs in the 3D reporter assay. The changes in the relative hTERT expression in MCF7 cells after drug treatment at two different time points (P1 and P2) were found to be the best variables to use in the logistic regression to develop a categorical prediction model, as follows:M = α_0_ + α_1_P1 + α_2_P2, where α_0_ = −0.65, α_1_ = −21.4, α_2_ = 67.3
with M < 0 for hTERT repressors and M > 0 for non-repressors. P2 is the average of the relative hTERT expression of the last two readings after drug addition minus P0, P1 is the average of the relative hTERT expression of the two readings before the last one after drug addition minus P0, and P0 is the average of the relative hTERT expression in the period before drug addition (0, 24, and 48 h). P1 and P2 represent the changes in hTERT expression after drug addition at two different time points to account for time-dependent fluctuations in the drug’s effects on cytotoxicity and hTERT expression in the 3D cultures. The logistic regression of two variables (P1 and P2) from the training dataset (pristimerin, blasticidin, cisplatin, doxorubicin, and paclitaxel) is illustrated in Figure 8. Also shown in the figure is tamoxifen, which was incorrectly classified as a non-repressor for hTERT in the model. As discussed earlier, tamoxifen should be an hTERT repressor based on its effect in terms of reducing the intracellular hTERT mRNA level. It would be necessary to further investigate tamoxifen for its possible roles in down- and/or upregulating hTERT expression with the 3000 bp hTERT promoter that includes the two estrogen-responsive elements [47].

## 4. Discussion

In this study, we aimed to develop a cell-based fluorescent reporter assay for the screening of drugs inhibiting hTERT expression in cancer cells. The transcriptional regulation of hTERT expression at the promoter region is complex. There are multiple levels of signaling pathways managing numerous positive and negative factors that influence TERT expression. To date, the known activators include, but are not limited to, c-Myc, NF-κB, STAT3, STAT5, and Pax, while the repressors include, but are not limited to, Mad 1, Sp3, CFTF, E2F1, and others. Some regulators, such as Sp1, AP-1, EGR-1, HIF-2, KLFs, and NFX1, play both roles [48,49]. The cells used to develop the hTERT repressor assay were MCF7 breast cancer cells, which have been widely used in drug discovery research [22,24] and in the investigation of hTERT regulation in cancer cells [50,51]. Padmanabhan et al. (2014) studied the hTERT mRNA expression level, telomerase activity, and telomere length in MCF7 cells using RT-PCR, the telomeric repeat amplification protocol (TRAP) assay, and terminal restriction fragment (TRF) southern blotting, respectively, and proved that the MCF7 cells had a high expression level of hTERT. Additionally, they connected the hTERT promoter fragments with a monomeric red fluorescent protein to visualize the promoter activity of hTERT in MCF7 cells [51]. Lamb et al. (2015) also examined the expression of hTERT in MCF7 cells using the green fluorescent protein GFP. They reported that fewer than 0.01% of cells in the stably transfected MCF7 cell pool had strong green fluorescence, which was also observed in our study. Additionally, they studied the proteomics related to mitochondrial function in these highly green-positive cells and demonstrated that hTERT-active cells exhibited stem-like, mitochondrial-rich characteristics in the heterogenous MCF7 population [50].

High-EGFP-expression cells were required for sensitive detection in the drug screening assay. In our study, we used the hTERT promoter from A549 lung cancer cells to drive the expression of EGFP in MCF7 cells because high levels of hTERT expression were observed in A549 cells in hTERT promoter-induced luciferase assays [52,53]. We cloned both 1500 bp and 330 bp of the hTERT promoter sequence from the genomes of A549 cells and compared the different promoter lengths for their effectiveness in driving EGFP expression. The transfection results showed no significant difference in the green fluorescence intensities between the 1500 bp and 330 bp promoters (Figure 1A). Therefore, only 330 bp promoter cells were processed for stable reporter cell generation. In addition to MCF7 cells, we also transfected A549 cells and Panc-1 pancreatic cancer cells with the EGFP expression plasmids and observed a larger proportion of green-positive Panc-1 cells than MCF7 and A549 cells (see Appendix A). To determine whether the engineered reporter cells were capable of responding correctly and sensitively to hTERT inhibitors, the engineered MCF7, A549, and PANC1 cells were treated with three hTERT inhibitors (pristimerin, EGCG, and BP) in 2D cultures, which showed that the hTERT gene expression was suppressed by more than 50% in MCF7 and A549 cells, but only moderately in PANC1 cells (see Appendix A). The EGFP expression controlled by the 330 bp hTERT promoter was also suppressed by these drugs in MCF7 cells, but not in A549 and PANC1 cells. Therefore, only engineered MCF7 cells were used to develop the 3D reporter assay for hTERT repressor screening.

The MCF7 reporter cells were cultured in 3D PET scaffolds to evaluate the drug effects on hTERT expression, because 3D cultures can better mimic the natural tumor physiology and provide more in vivo-like responses to drugs than 2D cultures [54]. We first demonstrated the 3D reporter assay with pristimerin and then further tested it with five additional drugs (doxorubicin, cisplatin, paclitaxel, blasticidin, and tamoxifen) with distinct mechanisms of action against cancer cells. The 3D reporter assay showed that pristimerin, blasticidin, cisplatin, and doxorubicin downregulated hTERT expression, while paclitaxel had negligible impact on hTERT (Figure 4, Figure 5 and Figure 6). These results were validated with the hTERT mRNA levels detected in qRT-PCR (Figure 7). Prior studies also showed that doxorubicin decreased hTERT mRNA in MCF7 and sarcoma cells [55,56] and paclitaxel had no impact on hTERT [42,43]. However, varied effects were reported in prior studies for cisplatin, which decreased the hTERT level in head and neck squamous cell carcinoma cells [57] but increased hTERT at lower concentrations in SMMC7721 human hepatocellular carcinoma cells [58] and had no significant effect in BEL-7404 human hepatoma cells [59]. Thus, a drug’s effect on hTERT expression could be cell-line-dependent as well as dose-dependent. For tamoxifen, we obtained inconsistent results in the 3D assay and mRNA test using cells from 2D cultures. The discrepancy in tamoxifen’s effect on hTERT in the 3D assay and the 2D mRNA test might be the result of the different culture environments. In general, cells cultured in 3D have higher tolerance and resistance to drugs and may respond to drugs differently compared to cells in 2D [21,23]. The quantification of mRNA in cells from 3D cultures may shed light on the cause.

The reduction in the relative hTERT expression caused by the drug’s inhibitory effect on the hTERT promoter was in the range of 10~20% in the 3D reporter assay. For this reason, we applied logistic regression to the 3D assay data to develop a predictive model for the classification of drugs into two categories—hTERT repressors and non-repressors (Figure 8). Logistic regression provides a reliable statistical method with the possibility to classify data into different categories [60] and has been widely used in risk assessment and medical research. We found that changes in the relative hTERT expression at two different time points after drug addition in the 3D assay could be used as input variables in binary logistic regression to develop a categorical model to predict hTERT repressors, which further validates the use of the 3D reporter assay developed in this study.

In the present reporter assay, the EGFP expression was driven by CMV and hTERT promoters in two different cells, respectively, cultured in different wells, which showed large variation and required five replicates in order to obtain statistically meaningful results. Because the two reporting constructs were engineered and randomly integrated into two different cells, the two resulting reporter cells were not biologically identical beside the reporting constructs, which might result in differences in cell growth and drug responses. In this study, we sought to identify CMV and hTERT cells that grew similarly to one another and maintained synchronous cell growth. However, the two cell types were cultured in different wells, and their responses to drug treatment, particularly at the late culture stage, might result in large variations. Alternatively, a dual-fluorescence reporter system with a CMV promoter-induced RFP as an internal control to correct the cell numbers could be used to increase the reproducibility with minimal experimental replicates, since cells expressing EGFP driven by the hTERT promoter and RFP driven by the CMV promoter can be cultured together in the same well [22,24]. Such a dual-fluorescence reporter assay achieved a greater reduction of 30~40% in the specific survivin expression level [24]. Therefore, it would be of great interest to develop a dual-reporter assay for hTERT repressor screening.

In this work, we observed that the hTERT promoter was significantly weaker than the CMV promoter, resulting in much lower fluorescence readings from the small number of EGFP transcripts initiated (Figure 1B). The low cell fluorescence brought about the challenge of sensitive and accurate detection, which was partially alleviated in the 3D culture, which intensified the fluorescence signal for easier detection [21]. A stronger fluorescent signal may be obtained by modifying the hTERT promoter by adding an enhancer element to the promoter region. Such research was conducted in the field of gene therapy, using hTERT as a cancer-specific promoter to induce a suicide gene to kill cancer cells [61]. A synthetic promoter (hTC) was created by combining the hTERT promoter and a minimal CMV promoter, which was capable of inducing high levels of gene expression while also exhibiting high cancer selectivity, as it was reported to be non-toxic to hTERT-negative human fibroblasts [62]. However, later research indicated that the hybrid promoter’s selectivity was lower than that of the native hTERT promoter [63]. Nevertheless, it may be worthwhile to use it as a reporter promoter and evaluate the reportability of the drug effect on the hTERT promoter.

It should be noted that the reporter assay used stable transformants generated by random or illegitimate integration to the genome. Locus-specific effects thus might have been responsible for the complex reactions that varied among the cells. Although selecting gene insertion sites, minimizing off-target insertion, and optimizing the integration efficiency are challenging, utilizing homologous site-specific integration holds great promise for predictable results [64]. However, illegitimate integration was easy and fast, and the cells employ illegitimate integration 1000–10,000 times more frequently than targeted integration [65,66]. In comparison to the technical difficulties associated with target integration, the disadvantage of random integration is the high workload for clone selection. The characterization of the gene locus using genome walking in conjunction with next-generation sequencing technology [67] may aid in our understanding of the locus-specific effects and resolution of the complex reporting cassette responses to drugs.

It is also worth noting that, apart from the transcriptional regulation of the hTERT promoter in cancers, mutations in the promoter region were also found to be strongly associated with hTERT’s reactivation in multiple malignant cancer cells [68]. Additionally, hypermethylation is another mechanism that increases the hTERT promoter’s activity [69]. The promoter fragments used to construct the hTERT reporting cassettes in this study were unmutated hTERT promoters. Cloning mutated hTERT promoters for drug screening or testing hypermethylation regulators could significantly broaden the 3D assay’s screening spectrum.

## 5. Conclusions

In summary, we developed a 3D cancer model that specifically reported hTERT expression regulation and demonstrated the feasibility of using the common fluorescent protein EGFP to indicate hTERT promoter activity for hTERT repressor screening. The assay was able to accurately indicate known and potential hTERT repressors in most conditions (more than 80%) evaluated. A predictive model was established via the logistic regression of the assay data, which can be used to identify and classify drugs as hTERT repressors or non-repressors. Additional drugs should be studied in the future to further improve and validate the assay and the logistic regression model.

## Figures and Tables

**Figure 1 bioengineering-12-00335-f001:**
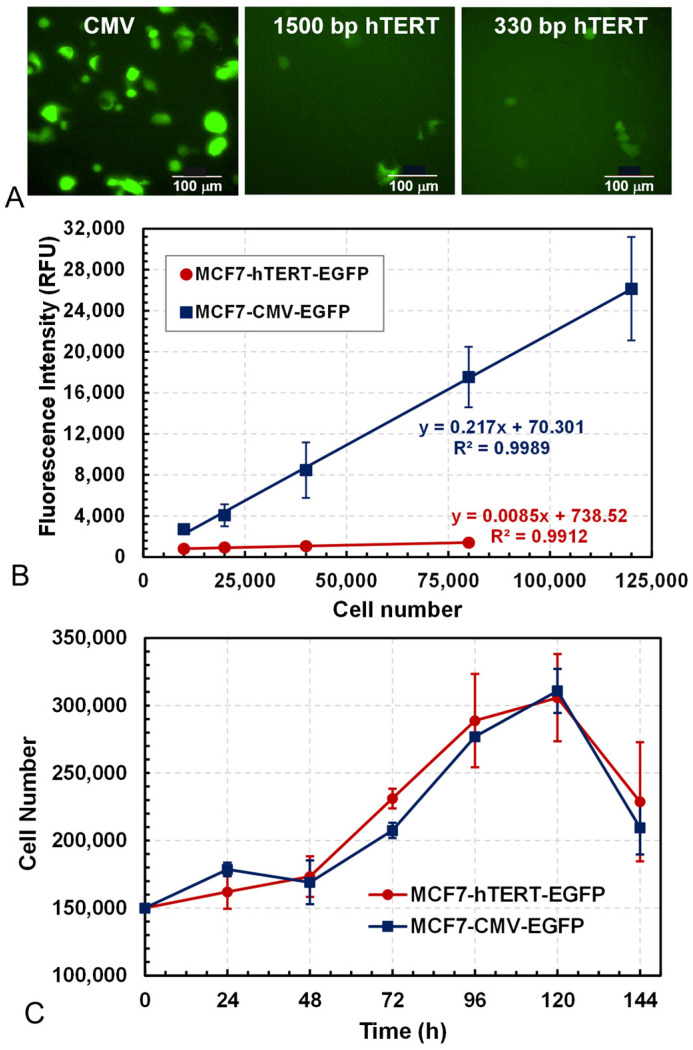
Characterization of fluorescent reporting cell lines MCF7-CMV-EGFP and MCF7-hTERT-EGFP. (**A**) Fluorescent microscopic images of MCF7 fluorescent reporter cells with EGFP expression controlled by CMV, 1500 bp hTERT, and 330 bp hTERT promoters, respectively. Images were taken 24 h after transfection at 480 ± 30 nm excitation and 535 ± 40 nm emission (scale bar: 100 µm). (**B**) Linear correlation between the EGFP fluorescence intensity (RFU) and cell number (R^2^ > 0.99). (**C**) Growth kinetics of MCF7-CMV-EGFP and MCF7-hTERT-EGFP cells in multi-wells.

**Figure 2 bioengineering-12-00335-f002:**
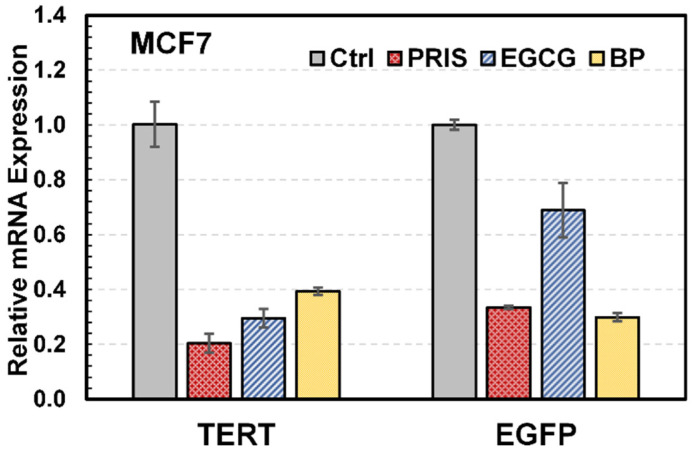
mRNA quantification of EGFP and hTERT expression levels in MCF7-hTERT-EGFP cells before and after treatment with 1.5 µM pristimerin (PRIS), 200 µM epigallocatechin-3-gallate (EGCG), or 80 µg/mL n-butylidenephthalide (BP) for 48 h. Ctrl: no drug.

**Figure 3 bioengineering-12-00335-f003:**
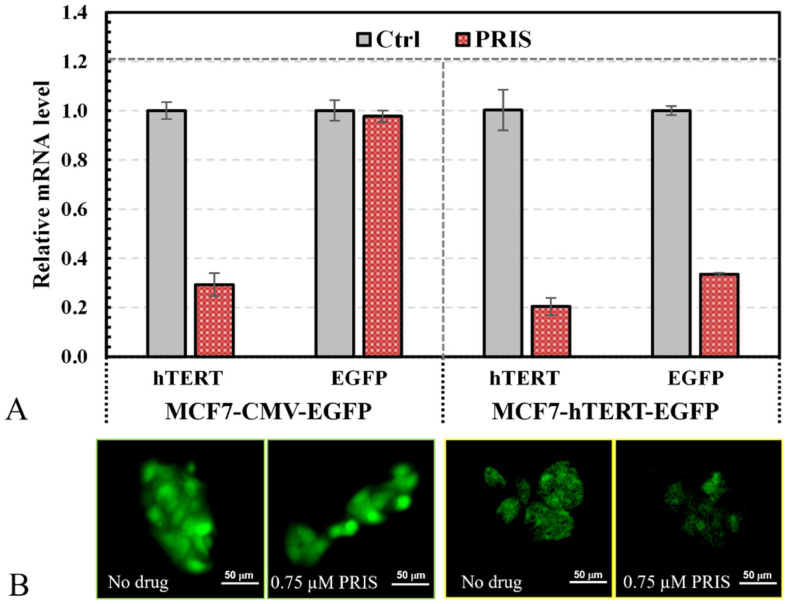
Characterization of MCF7-hTERT-EGFP and MCF7-CMV-EGFP reporting cell pair. (**A**) mRNA quantification of EGFP and hTERT in MCF7-CMV-EGFP (**left**) and MCF7-hTERT-EGFP (**right**) cells before (Ctrl, no drug) and after treatment with 1.5 µM pristimerin (PRIS) for 48 h. (**B**) Cell fluorescence images before and after treatment with 0.75 µM pristimerin for 96 h. Cell nuclei were stained blue to visualize the cell status and cell locations (scale bar: 50 µm).

**Figure 4 bioengineering-12-00335-f004:**
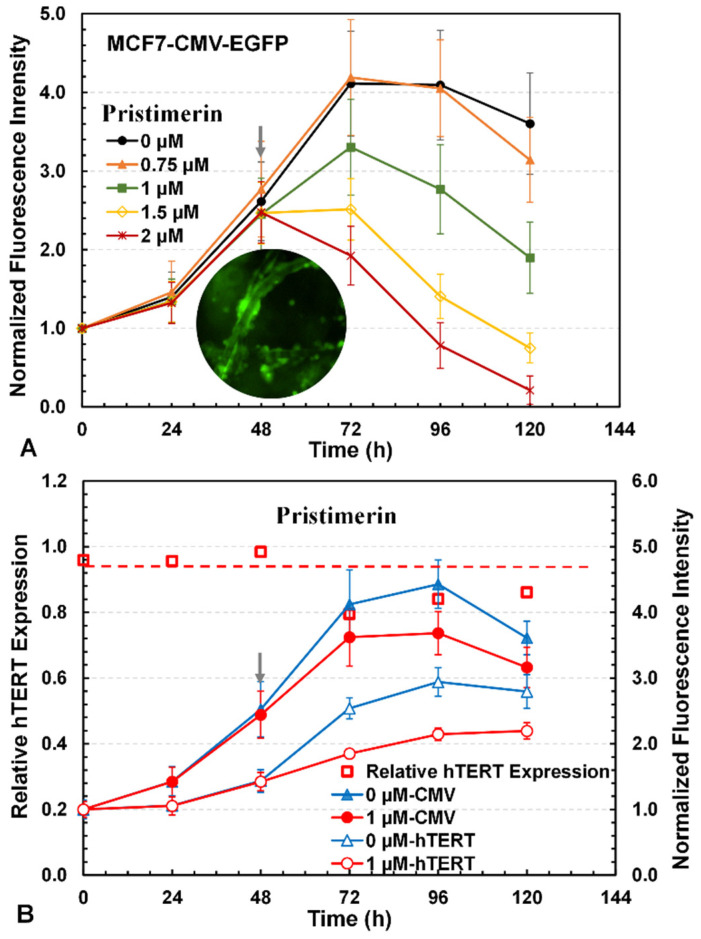
Effects of pristimerin on 3D MCF7 cultures. (**A**) Dose-dependent cytotoxicity as indicated by changes in green fluorescence from MCF7-CMV-EGFP cells exposed to different concentrations of pristimerin. The photo shows the fluorescent cells in the 3D culture. (**B**) Changes in normalized green fluorescence intensities, with the initial value as one, from MCF7-hTERT-EGFP and MCF7-CMV-EGFP cells and the relative hTERT expression level in MCF7 cells treated with pristimerin at 1 µM. Relative hTERT expression = [NF-hTERT/NF-CMV]_with drug_/[NF-hTERT/NF-CMV]_without drug_, where NF-hTERT and NF-CMV are the normalized green fluorescence intensities of MCF7-hTERT-EGFP and MCF7-CMV-EGFP, respectively.

**Figure 5 bioengineering-12-00335-f005:**
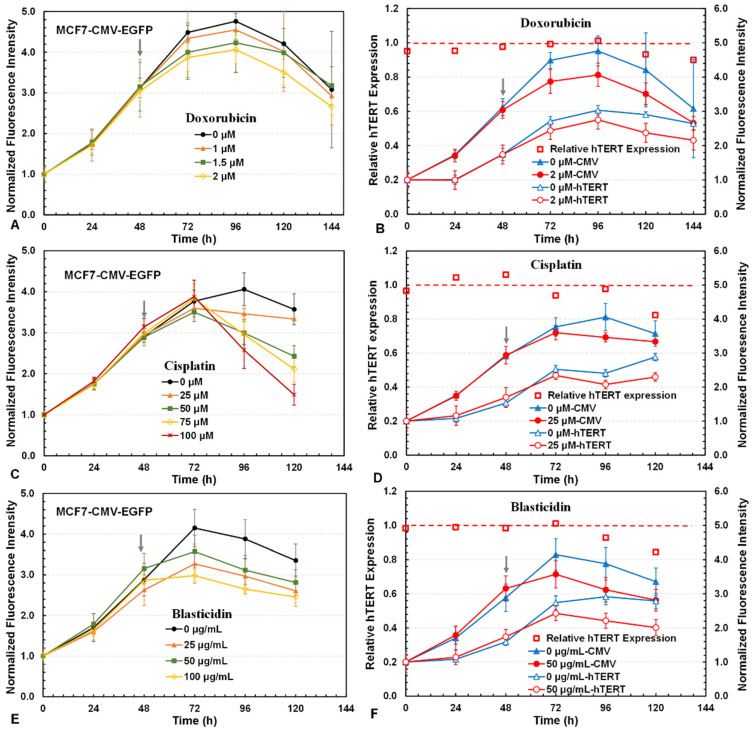
Three-dimensional reporter assays for drug cytotoxicity and hTERT expression by doxorubicin, cisplatin, and blasticidin. Cytotoxicity indicated by the changes in green fluorescence from MCF7-CMV-EGFP cells for (**A**) doxorubicin, (**C**) cisplatin, and (**E**) blasticidin. Changes in normalized green fluorescence intensities in MCF7-hTERT-EGFP and MCF7-CMV-EGFP cells and the relative hTERT expression levels in MCF7 cells treated with (**B**) 2 µM doxorubicin, (**D**) 25 µM cisplatin, and (**F**) 50 µg/mL blasticidin.

**Figure 6 bioengineering-12-00335-f006:**
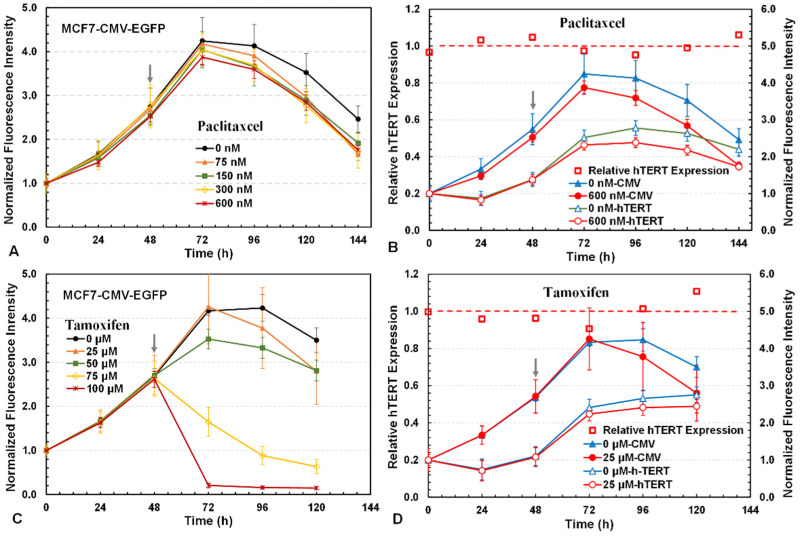
Three-dimensional reporter assays for drug cytotoxicity and hTERT expression by paclitaxel and tamoxifen. Cytotoxicity indicated by the changes in green fluorescence from MCF7-CMV-EGFP cells for (**A**) paclitaxel and (**C**) tamoxifen. Changes in normalized green fluorescence intensities in MCF7-hTERT-EGFP and MCF7-CMV-EGFP cells and the relative hTERT expression levels in MCF7 cells treated with (**B**) 600 nM paclitaxel and (**D**) 25 µM tamoxifen.

**Figure 7 bioengineering-12-00335-f007:**
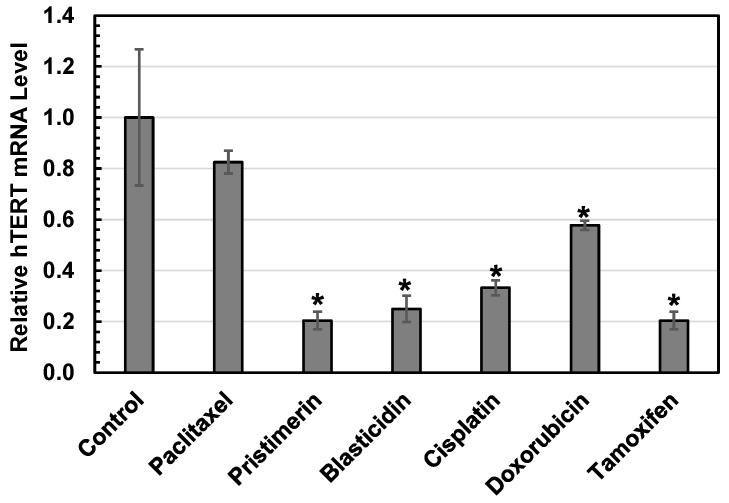
mRNA levels of hTERT in MCF7 cells after treatment with various anticancer drugs. The tested drug concentrations were 600 nM for paclitaxel, 1.5 µM pristimerin, 50 µg/mL for blasticidin, 25 µM for cisplatin, 2 µM for doxorubicin, and 25 µM for tamoxifen. Data shown are the averages from triplicate samples, with the error bar indicating the standard error. * indicates a significant difference from the control (no drug treatment). Except for paclitaxel, all drugs significantly decreased the hTERT mRNA level, confirming their downregulation effects on hTERT in MCF7 cells.

**Figure 8 bioengineering-12-00335-f008:**
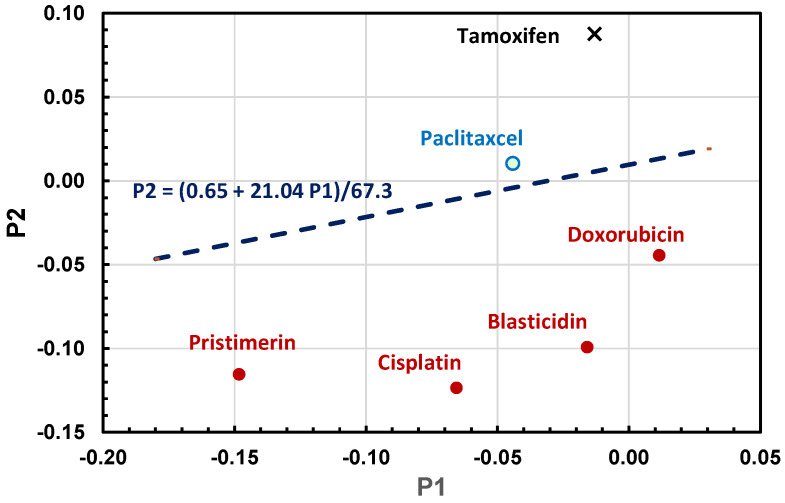
Logistic regression of changes in the relative expression of hTERT (P1 and P2) in 3D cultures of MCF7 cells treated with 1 µM pristimerin, 2 µM doxorubicin, 25 µM cisplatin, 600 nM paclitaxel, 50 µg/mL blasticidin, and 25 µM tamoxifen, respectively. The dotted line shows the prediction model separating hTERT repressors (red circles, below the line) and non-repressors (blue circle, above the line) used in the training dataset.

**Table 1 bioengineering-12-00335-t001:** Relative hTERT expression in MCF7 cells treated with various drugs in the 3D reporter assay and the variables used in the logistic regression for a categorical prediction model.

	0 h	24 h	48 h	72 h	96 h	120 h	144 h	P0	P1	P2	M
Blasticidin	0.98	0.99	0.98	1.01	0.93	0.84	-	0.99	−0.016	−0.099	−6.992
Cisplatin	0.97	1.04	1.06	0.94	0.98	0.82	-	1.02	−0.066	−0.123	−7.573
Pristimerin	0.96	0.96	0.98	0.80	0.84	0.86	-	0.97	−0.148	−0.115	−5.295
Doxorubicin	0.95	0.95	0.98	0.99	1.01	0.93	0.90	0.96	0.012	−0.044	−3.877
Paclitaxcel	0.97	1.03	1.05	0.97	0.95	0.99	1.06	1.01	−0.044	0.010	0.973
Tamoxifen	1.00	0.96	0.96	0.91	1.01	1.11	-	0.97	−0.013	0.088	5.522

M = a_0_ + a_1_P1 + a_2_P2 [a_0_ = −0.65, a_1_ = −21.4, a_2_ = 67.3]. P0 = Average of the relative hTERT expression in the period before drug addition (0, 24, and 48 h). P1 = Average of the relative hTERT expression of the last two readings after drug addition—P0. P2 = Average of the relative hTERT expression of the two readings before the last one after drug addition—P0.

## Data Availability

The data presented in this study are available upon request to the corresponding author.

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
