# Peer review of "Development of 3D Cell-Based Fluorescent Reporter Assay for Screening of Drugs Downregulating Telomerase Reverse Transcriptase"

_bioengineering, 2025, doi:10.3390/bioengineering12040335_

Round 1

Reviewer 1 Report

Comments and Suggestions for Authors

The paper titled "Development of 3D Cell-Based Fluorescent Reporter Assay for Screening Drugs Down-Regulating Telomerase Reverse Transcriptase," presents by Yang et al. reported a novel 3D cell-based fluorescent reporter assay for identifying hTERT (human telomerase reverse transcriptase) inhibitors. The experimental design is reasonable and relatively complete, and the data support the main conclusions, demonstrating the potential of this assay for high-throughput screening of anticancer drugs and making a meaningful contribution to the field of cancer treatment. I therefore recommend acceptance of this manuscript in Bioengineering.

Several errors in the manuscript are listed as below:

  1. The scale bar in Fig 1A is unclear, and the scale bar in Fig 3B is missing.
  2. The authors should explain the inconsistent results for tamoxifen between the 3D assay and mRNA tests.
  3. The assay is validated predominately in MCF7 cells, testing in additional cell lines could improve its generalizability.
  4. The authors should adjust/improve the figure and table size of figure 3, 5-7 and table 1 according to the page.

Author Response

Comment 1. The scale bar in Fig 1A is unclear, and the scale bar in Fig 3B is missing.

Responses – We have corrected with clearer scale bar in Fig. 1A and Fig. 3B.

Comment 2. The authors should explain the inconsistent results for tamoxifen between the 3D assay and mRNA tests.

Responses – This has been discussed in the manuscript (Section 3.5, page 16) “Previous studies reported that treatment of MCF7 cells with tamoxifen resulted in hTERT repression and cell inhibition [45, 46]. Depending on the cell type, tamoxifen could inhibit or promote cell growth and repress or activate hTERT, possibly resulting from its dual-role regulation on the two estrogen responsive elements (ERE) on the hTERT promoter. The EREs are located at around -2700 and -970 upstream of the promoter region [47], which were not present in the (330 bp)-hTERT-EGFP reporting cassettes used in the 3D reporter assay. This might contribute to the discrepancy in the results from the 3D reporter assay and mRNA test.” And “It would be necessary to further investigate the tamoxifen for its possible roles in down- and/or up-regulating hTERT expression with the 3000 bp hTERT promoter that includes the two estrogen responsive elements [47]. We added “Another possible reason could be the difference in the culture environment used in the 3D reporter assay (3D culture) versus the 2D culture for incubating the cells collected for mRNA assay. It is well recognized that 3D cultures can better represent the real tumor microenvironment, including cell-cell and cell-matrix interactions, as well as their effects on cell responses to drugs, and thus give drug toxicity results that are closer to in vivo [19, 20]. This remains to be investigated in the future.” in the revised manuscript. In fact, we have already discussed this point in Discussion (page 20), “The discrepancy in tamoxifen effect on hTERT in the 3D assay and the 2D mRNA test might be the results of different culture environments. In general, cells cultured in 3D have higher tolerance and resistance to drugs and may respond to drugs differently compared to cells in 2D [21, 23]. Quantification of mRNA in cells from 3D cultures may shed light on the cause.”

Comment 3. The assay is validated predominately in MCF7 cells, testing in additional cell lines could improve its generalizability.

Responses – We did test two other cancer cell lines. As already discussed in the Discussion (page 19), “In addition to MCF7 cells, we have also transfected A549 cells and Panc-1 pancreatic cancer cells with the EGFP expression plasmids and observed a higher proportion of green positive Panc-1 cells than MCF7 and A549 cells (see Figure S1). To determine whether the engineered reporter cells were capable of responding correctly and sensitively to hTERT inhibitors, the engineered MCF7, A549, and PANC1 cells were treated with three hTERT inhibitors (pristimerin, EGCG, and BP) in 2D cultures, which showed that the hTERT gene expression was suppressed by more than 50% in MCF7 and A549 cells, but only moderately in PANC1 cells (see Figure S2). The EGFP expression controlled by the 330 bp hTERT promoter was also suppressed by these drugs in MCF7 cells, but not in A549 and PANC1 cells. Therefore, only engineered MCF7 cells were used to develop the 3D reporter assay for hTERT repressor screening.” Because of the inconsistent results from A549 and PANC1 cells we obtained, we only focused on MCF7 in this study to demonstrate the feasibility of the 3D TERT reporter assay. 

Comment 4. The authors should adjust/improve the figure and table size of figure 3, 5-7 and table 1 according to the page.

Responses – We have adjusted the figure and table size of Figures 3, 5-7 and Table 1 to fit for the page.

Reviewer 2 Report

Comments and Suggestions for Authors

This study has developed and demonstrated the efficacy of a 3D fluorescent reporter assay for screening drugs that suppress hTERT expression. Using EGFP as an indicator, the study allows real-time visualization of hTERT promoter activity, providing a faster and more convenient alternative to conventional mRNA quantification methods. In addition, the incorporation of a 3D culture environment aims to evaluate drug effects under conditions that more closely resemble the tumor microenvironment. If successful, this approach has the potential to contribute to the analysis of hTERT regulatory mechanisms and applications in drug discovery, making it a valuable tool in the development of cancer therapies. However, several concerns remain, and this reviewer would like to seek the authors' perspective on the following points.

(1) In this study, the reporter genes (hTERT-EGFP and CMV-EGFP) were randomly integrated into the genome, which may lead to variations in promoter activity and differences between cell clones depending on the integration site. In particular, during the screening process, selection of drug responsive cells based on hTERT promoter activity may result in preferential survival of clones with mutations in the promoter region. This raises concerns that the assessment may not accurately reflect the natural expression of hTERT.

(2) The study does not fully clarify whether the observed decrease in hTERT promoter activity is due to specific inhibition of hTERT expression or to non-specific effects such as cytotoxicity or reduced cell proliferation. In particular, high concentrations of pristimerin, cisplatin, and blasticidin may induce cell death in addition to inhibiting hTERT, so it is critical to establish a clear correlation between hTERT suppression and cell viability. The authors appear to have attempted to address this issue by comparing GFP expression under the control of the CMV promoter; however, the CMV promoter is known to be susceptible to environmental factors and may introduce non-specific variability.

(3) The study suggests that the use of a 3D culture system may better mimic the tumor microenvironment and provide a more in vivo reflection of hTERT regulation. However, a detailed comparison between 2D and 3D cultures regarding hTERT expression levels, cell proliferation rates, and drug penetration effects was not performed. In particular, differences in cell-cell interactions and nutrient distribution in 3D cultures could influence hTERT expression and drug response. Without verification of these parameters, it is difficult to confirm that the 3D assay provides a truly in vivo assessment.

Author Response

Comment 1. In this study, the reporter genes (hTERT-EGFP and CMV-EGFP) were randomly integrated into the genome, which may lead to variations in promoter activity and differences between cell clones depending on the integration site. In particular, during the screening process, selection of drug responsive cells based on hTERT promoter activity may result in preferential survival of clones with mutations in the promoter region. This raises concerns that the assessment may not accurately reflect the natural expression of hTERT.

Responses – As mentioned on page 23, “The promoter fragments used to construct the hTERT reporting cassettes in this study were unmutated hTERT promoter.” We already considered downside of random integration of the promoter gene on the genome, that’s why we verified the engineered MCF7 cells with three known HTERT inhibitors to make sure the selected cells responded correctly. We used G418 to screen stale cells. G418 should not impact the HTERT promoter region, so there should be no mutation from this process. After this, when we selected cells that responded to hTERT inhibitors, the cells were only treated with the inhibitors for a few days. So the likelihood of hTERT mutation during this short exposure time would be very low, even though there was a mutation from the exposure. The cells were not reused for subsequent drug tests. Therefore, the assessment with the reporter assay should reflect the natural expression of hTERT as also verified with mRNA expression measurements.

Comment 2. The study does not fully clarify whether the observed decrease in hTERT promoter activity is due to specific inhibition of hTERT expression or to non-specific effects such as cytotoxicity or reduced cell proliferation. In particular, high concentrations of pristimerin, cisplatin, and blasticidin may induce cell death in addition to inhibiting hTERT, so it is critical to establish a clear correlation between hTERT suppression and cell viability. The authors appear to have attempted to address this issue by comparing GFP expression under the control of the CMV promoter; however, the CMV promoter is known to be susceptible to environmental factors and may introduce non-specific variability.

Responses – We addressed this cytotoxicity effect by using the CMV cells as the control. This is already discussed in great detail in section 3.3 (page 14). “We investigated the down-regulation effect of pristimerin at 1 µM on hTERT in the 3D cultures of MCF7-hTERT-EGFP and MCF7-CMV-EGFP cells.” “To evaluate the drug effect on down-regulating the hTERT promoter, the relative hTERT expression was estimated from [NF-hTERT/NF-CMV]with drug/[NF-hTERT/NF-CMV]without drug, where NF-hTERT is the normalized green fluorescence intensity of MCF7-hTERT-EGFP and NF-CMV is the normalized green fluorescence intensity of MCF7-CMV-EGFP. As shown in Fig. 4B, the relative hTERT expression was approximately one in the absence of the drug but decreased significantly to below one after drug addition due to the down-regulation of the hTERT promoter by pristimerin, which was confirmed with the mRNA level of hTERT in MCF7 cells with and without drug treatment (Fig. 3).”

Comment 3. The study suggests that the use of a 3D culture system may better mimic the tumor microenvironment and provide a more in vivo reflection of hTERT regulation. However, a detailed comparison between 2D and 3D cultures regarding hTERT expression levels, cell proliferation rates, and drug penetration effects was not performed. In particular, differences in cell-cell interactions and nutrient distribution in 3D cultures could influence hTERT expression and drug response. Without verification of these parameters, it is difficult to confirm that the 3D assay provides a truly in vivo assessment.

Responses – As discussed on page 11, “The growth kinetics of MCF7-CMV-EGFP and MCF7-hTERT-EGFP cells in 6-well plate cultures were monitored by counting the cell number daily. Figure 1C shows that MCF7-CMV-EGFP and MCF7-hTERT-EGFP cells had similar growth kinetics, indicating that the EGFP expression did not significantly affect cell growth behavior.” Therefore, the EGFP can be used to monitor cell growth in 3D cultures as well, which has been verified in our prior studies. “The application of 3D culture improves drug toxicity prediction for drug screening assays. In general, 3D cultures can better represent the real tumor microenvironment, including cell-cell and cell-matrix interactions, as well as their effects on cell responses to drugs, and thus give drug toxicity results that are closer to in vivo [19, 20]. In our previous work, we developed a polyethylene terephthalate (PET)-based 3D culture and showed that cells growing on the PET scaffold exhibited in vivo-like morphology [21]. The scaffold embedded in a modified 384-well plate allowed for easy detection of red and green fluorescent protein from engineered reporter cells using a plate reader. We also found that this 3D culture greatly improved fluorescent signal readings due to the higher cell culture density and the 3D scaffold’s fluorescence focusing effect. This 3D assay has been successfully applied for the screening of survivin gene inhibitors [22–24].” A detailed comparison between 2D and 3D cultures regarding hTERT expression levels, cell proliferation rates, and drug penetration effects would be desirable but not really necessary for the purpose of the present study. Moreover, it is not possible to compare the assays in 2D and 3D because the 2D culture had very weak EGFP fluorescence that would not be possible to use in the assay.

Reviewer 3 Report

Comments and Suggestions for Authors

The authors selected an important target, human telomerase reverse transcriptase which is a catalytic subunit of the enzyme telomerase and could be a potential target gene for cancer therapy. The main idea of the study is developing a cell-based fluorescent reporter assay for screening drugs inhibiting hTERT expression in cancer cells. The research demonstrates that hTERT promoter with EGFP as a reporter can be applied for screening potential cancer drugs targeting hTERT.

Sufficient information on hTERT as a therapeutic target for cancer drug discovery and development is provided. In the introduction part authors also explained why the application of 3D culture improves drug toxicity prediction for drug screening approaches.

All experimental issues are presented well, the steps of research are clear and informative.

Strong green fluorescence was recorded for cells transfected with CMV-EGFP plasmid. Figures are of good quality; validation of Reporting Cell Pair is written in detail. The important finding was that according to fluorescent microscopy images 0.75 mM pristimerin did not affect EGFP expression in MCF7-CMV-EGFP cells but significantly decreased EGFP expression in MCF7-hTERT-EGFP cells. 3D reporter assays were applied for measuring the cytotoxicity of three drugs and hTERT expression. Further improvement and validation of the assay can become the subject of further research.

The manuscript is attractive for a wide audience and can be published in the current form.

Author Response

The authors selected an important target, human telomerase reverse transcriptase which is a catalytic subunit of the enzyme telomerase and could be a potential target gene for cancer therapy. The main idea of the study is developing a cell-based fluorescent reporter assay for screening drugs inhibiting hTERT expression in cancer cells. The research demonstrates that hTERT promoter with EGFP as a reporter can be applied for screening potential cancer drugs targeting hTERT.

Sufficient information on hTERT as a therapeutic target for cancer drug discovery and development is provided. In the introduction part authors also explained why the application of 3D culture improves drug toxicity prediction for drug screening approaches.

All experimental issues are presented well, the steps of research are clear and informative.

Strong green fluorescence was recorded for cells transfected with CMV-EGFP plasmid. Figures are of good quality; validation of Reporting Cell Pair is written in detail. The important finding was that according to fluorescent microscopy images 0.75 mM pristimerin did not affect EGFP expression in MCF7-CMV-EGFP cells but significantly decreased EGFP expression in MCF7-hTERT-EGFP cells. 3D reporter assays were applied for measuring the cytotoxicity of three drugs and hTERT expression. Further improvement and validation of the assay can become the subject of further research.

The manuscript is attractive for a wide audience and can be published in the current form.

Response - Thanks for the reviewer's all positive assessment of our work.

Round 2

Reviewer 2 Report

Comments and Suggestions for Authors I am in agreement with the acceptance of this paper as the authors have adequately addressed the concerns of my peer review.